New insight into the divergent responses of plants to warming in the context of root endophytic bacterial and fungal communities

Wei Xiaoting 1
Jiang Fengyan 1
Han Bing 1
Zhang Hui 1
Huang Ding huangding@cau.edu.cn 1
Shao Xinqing 1 2 3
1 College of Grassland Science and Technology, China Agricultural University , Beijing , China
2 Key Laboratory of Restoration Ecology of Cold Area in Qinghai province, Northwest Institute of Plateau Biology, Chinese Academy of Sciences , Xining , China
3 Qinghai Provincial Key Laboratory of Adaptive Management on Alpine Grassland , Xining , China
Kalendar Ruslan
Electronic publication date: 2021 May 26
Publication date: 2021
Volume: 9
Electronic Location ID: e11340
Received 2021 Jan 6; Accepted 2021 Apr 3
Copyright: ©2021 Wei et al.
Copyright year: 2021
Copyright holder: Wei et al.
License: This is an open access article distributed under the terms of the Creative Commons Attribution License, which permits unrestricted use, distribution, reproduction and adaptation in any medium and for any purpose provided that it is properly attributed. For attribution, the original author(s), title, publication source (PeerJ) and either DOI or URL of the article must be cited.
License URL: https://creativecommons.org/licenses/by/4.0/

Keywords: Qinghai-Tibet Plateau, Elevation gradient, Climate warming, Root endophytic community, Kobresia pygmaea, Elymus nutans

Funding: Ministry of Science and Technology of China 2016YFC0501902 National Natural Science Foundation of China 31971746 Platform of Adaptive Management on Alpine Grassland-livestock System 2020-ZJ-T07 Key Scientific and Technological Special Projects of Qinghai Province 2018-NK-A2 This work was financially supported by the Ministry of Science and Technology of China [2016YFC0501902], the National Natural Science Foundation of China [31971746], Platform of Adaptive Management on Alpine Grassland-livestock System [2020-ZJ-T07] and the Key Scientific and Technological Special Projects of Qinghai Province [2018-NK-A2]. The funders had no role in study design, data collection and analysis, decision to publish, or preparation of the manuscript.

==============================
Plant adaptation under climate changes is critical to the maintenance of terrestrial ecosystem structure and function. Studying the response of the endophytic community to climate warming is a novel way to reveal the mechanism of host environmental adaptability because of the prominent role endophytes play in host nutrient acquisition and stress tolerance. However, host performance was generally neglected in previous relevant research, which limits our understanding of the relationships between the endophytic community and host responses to climate warming. The present study selected two plants with different responses to climate warming. Elymus nutans is more suitable for growing in warm environments at low altitude compared to Kobresia pygmaea. K. pygmaea and E. nutans were sampled along an altitude gradient in the natural grassland of Qinghai-Tibet Plateau, China. Root endophytic bacterial and fungal communities were analyzed using high throughput sequencing. The results revealed that hosts growing in more suitable habitats held higher endophytic fungal diversity. Elevation and host identity significantly affected the composition of the root endophytic bacterial and fungal community. 16S rRNA functional prediction demonstrated that hosts that adapted to lower temperatures recruited endophytic communities with higher abundance of genes related to cold resistance. Hosts that were more suitable for warmer and drier environments recruited endophytes with higher abundance of genes associated with nutrient absorption and oxidation resistance. We associated changes in the endophytic community with hosts adaptability to climate warming and suggested a synchronism of endophytic communities and hosts in environmental adaptation.

Introduction

Plant endophytes are microorganisms that colonize the intercellular space of plants and establish a harmonious association with their host plants (Odelade & Babalola, 2019). The genome of the host and its microbiome are considered a unit according to the hologenome theory (Zilber-Rosenberg & Rosenberg, 2008). Studies on plant-associated microbiota recently exhibited phenomenal expansion in the areas of natural and agricultural ecosystems due to their great potential in improving host fitness, including nutrient acquisition (Van der Heijden et al., 2016), stress tolerance (Liu et al., 2020a; Liu et al., 2020b) and disease defense (Dini-Andreote, 2020; Carrión et al., 2019). Endophytic community diversity was closely associated with multiple ecosystem functions (Laforest-Lapointe et al., 2017).

Recently, great attention has been paid to the environmental problems caused by climate warming in terrestrial ecosystems, such as the loss of biodiversity, changes in vegetation composition and community succession. Climate changes also affect plant ranges and allow some species to expand or narrow their ranges (Morin, Viner & Chuine, 2008). The formation of new interspecific relationships between plant species under climate change may drive these ecological processes (Little et al., 2016), and endophytes may modulate the interspecific interactions and vegetation composition (Porras-Alfaro & Bayman, 2011; Jackrel et al., 2020).

Plant endophytic communities are dynamic and variable among seasons (Li, Yang & Zhao, 2005) and environments (Laforest-Lapointe, Messier & Kembel, 2016) and reflect the adaptability of plants to changing environments. The recent “cry for help” hypothesis suggests that plants selectively recruit beneficial microbes from their surroundings under biotic or abiotic stress (Neal et al., 2012; Liu & Brettell, 2019). Pseudomonas colonization in the roots of the desert plant Alhagi sparsifolia increased under drought conditions and promoted drought resistance (Zhang et al., 2020). Different plant species show unique abilities in the process of recruitment. For example, most plants form symbioses with arbuscular mycorrhizal fungi (AMF) to increase phosphorus availability. Arabidopsis thalian, which lose the ability of symbiosing with AMF, recruit Helotiales and Colletotrichum tofieldiae under phosphorus-limited conditions (Almario et al., 2017; Hiruma et al., 2016).

However, studies on the impacts of climate change on plant endophyte communities are insufficient and many findings are conflicting. Endophytes and their hosts are “communities of interest”. Previous studies focused only on endophytic communities and host responses to climate change were generally neglected (Qian et al., 2018; Cordier et al., 2012), which limits our understanding of the ecological significance of endophytic community changes. Host selection is critical because plant species (Yao et al., 2019; Laforest-Lapointe, Messier & Kembel, 2016) and compartments (leaf, stem, root) of the same plant greatly affect endophytic community composition (Qian et al., 2019). Fujimura also concluded that the conflicting results between studies on the effects of warming on endophytic communities may due to differences in host responses (Fujimura, Egger & Henry, 2008). Therefore, plant species that are sensitive to climate changes should be selected to examine the responses of endophyte communities and improve our understanding of plant-microbe interactions and elucidate the mechanisms of plant adaptation to climate change from the perspective of endophytes.

The Qinghai-Tibet Plateau is the highest plateau in the word, and it is experiencing rapid climate warming at rate more than twice the global level (Ma et al., 2017), which threatens biodiversity and ecosystem function in this area. Kobresia pygmaea and Elymus nutans are two common plants in the Qinghai-Tibet Plateau that respond differently to climate warming. Kobresia pygmaea is a carpet-like sedge species that is highly adaptable to high-altitude environments, and it primarily grows on the alpine meadow of the Qinghai-Tibet Plateau between altitudes of 3,000 m and 6,000 m (Wu et al., 2017; Miehe et al., 2008). The K. pygmaea alpine meadow, which is dominated by K. pygmaea, covers 17% of the total area of the Qinghai-Tibet Plateau (Miehe et al., 2008), which makes it an important species in maintaining the stability of the local ecosystem. Elymus nutans is a dominant and constructive grass of meadow steppe, and it is characterized by strong drought, cold and pest resistance. It is widely distributed and survives at elevations ranging from 450 m to 4,500 m. E. nutans is better suited to low altitudes and warmer habitats than K. pygmaea, and an increase in temperature from high to low elevations promotes the growth of E. nutans (Qi et al., 2020). K. pygmaea primarily grows at high altitudes in humid and cold habitats (Wu et al., 2017). Warming increased the proportion of Elymus nutans and decreased the proportion of K. pygmaea in alpine grassland ecosystems (Liu et al., 2018; Niu et al., 2019).

The present study examined the adaptability of plants to climate warming by analyzing changes in endophytic community diversity and composition. We used elevation gradients where climatic conditions, vegetation and soil characteristics show regular changes over short geographic distances to investigate the responses of terrestrial ecosystems to climate warming (Korner, 2007). K. pygmaea and E. nutans were sampled along an elevation gradient in the Qinghai-Tibet Plateau, and the root endophytic bacterial and fungal communities were analyzed based on the high-throughput sequencing. The innovation of this study lies in the selection of the two hosts with opposing responses to climate warming to elucidate how endophytic communities change during the promotion or inhibition of host growth by climate warming. We addressed the following questions: (1) how endophytic community diversity in the roots of K. pygmaea and E. nutans changed with climate warming (decreasing elevation); (2) whether climate warming affected endophytic community composition in K. pygmaea and E. nutans; and (3) how changes in the endophytic community explain host responses to climate warming.

Materials and Methods

Study site and sampling

Fieldwork was performed in August 2019 in the middle of the Qilian Mountains (37.68°N, 100.75°E) near Qilian County, which is located in the northeastern part of Qinghai Province, China. With an average elevation over 3,000 m, this area experiences a typical plateau continental climate. The mean annual and growing season temperatures (from June to August) are 1 °C and 13 °C, respectively. The annual mean precipitation is 480 mm, with 80% occurring during the growing season. The climate and vegetation composition change significantly with elevation in the Qilian Mountains. We sampled at elevations of 3 350 m, 3,460 m, 3,570 m, 3,680 m, and 3,800 m above sea level (a.s.l.) on the south slope of the mountain. The temperature decreases and the precipitation increases with increasing altitude in this area (Jin et al., 2017; Li et al., 2018). According to the vegetation investigation, the grassland was classified as alpine meadows at elevations of 3,350 m, 3,460 m, and 3 570 m and alpine shrub meadows at elevations of 3,680 m and 3,800 m.

We defined three plots at each elevation, with each plot 20 m away from the other plots. The roots of K. pygmaea and E. nutans were sampled around each plot. K. pygmaea always aggregates in grasslands and forms slightly yellow green patches because of its dense network of roots, which are produced clonally. Therefore, patches of K. pygmaea and clumps of E. nutans were dug using a sterilized shovel from a depth of 10 cm. We reserved the belowground parts, including the soil and roots, when sampling, and separated the roots after transported to the laboratory. We reserved the belowground parts, including the soil and roots, when sampling, and separated the roots after transport to the laboratory. Five individuals of each plant species per plot composed one sample, which resulted in 30 root samples (2 species × 5 elevations × 3 replicates). All samples were placed in labeled sterile plastic bags and immediately stored in an incubator with dry ice. Samples were transported to the laboratory and stored at −80 °C until further processing. The roots of K. pygmaea and E. nutans were separated in the laboratory using sterilized scissors and brushes then transferred to 50-mL sterile centrifuge tubes containing 30 mL of 0.01 M sterilized PBS (136 mM NaCl, 8 mM Na2HPO4, 2 mM KH2PO4, 2.6 mM KCl, pH 7.2). The tubes were placed on a shaking platform (180 rpm) for 20 min to remove the soil attached to the root. The roots were rinsed in deionized water and surface sterilized according to the following steps: immersion in sterile water twice for 30 s, 75% ethanol for 1 min, 3.25% sodium hypochlorite for 3 min, 75% ethanol for 30 s, sterile water for 1 min, and a final rinse with sterile water. The surface-sterilized roots were stored in a freezer at −80 °C.

Genomic DNA extraction, amplification and purification

Endophytic genomic DNA was extracted from 0.5 g of frozen surface-sterilized roots using the MOBIO PowerSoil® DNA Isolation Kit (MOBIO Laboratories, Carlsbad, CA, USA). The integrity and purity of the extracted DNA were tested on 1% agarose gels, and a NanoDrop One was used to measure the concentration and purity of the extracted DNA. The 16S rRNA gene V5-V7 region was amplified using the primers 799F (5′-AACMGGATTAGATACCCKG-3′) and 1193R (5′-ACGTCATCCCCACCTTCC-3′) with a 12-bp barcode (Horton et al., 2014). The primer set 799F-1193R reduces co-amplification levels of mitochondrial gene during endophytic 16S rRNA gene PCR amplification (Wang et al., 2018). The internal transcribed spacer 1(ITS1) region of fungal rRNA was amplified using the forward primer 5′-CTTGGTCATTTAGAGGAAGTAA-3′  and the reverse primer 5′-GCTGCGTTCTTCATCGATGC-3′  (Horton et al., 2014). Primers were constructed by Invitrogen (Carlsbad, CA, USA). A 50-µL reaction containing 25 µL of 2×  PremixTaq (Takara Biotechnology, Dalian Co. Ltd., China), 1 µL of each primer (10 mM) and 3 µL of DNA template (20 ng/µL) was used for PCR amplification in a BioRad S1000 (Bio-Rad Laboratory, CA). The following thermal cycling process was used: 5 min at 94 °C for initialization; 30 cycles of denaturation at 94 °C for 30 s; annealing at 52 °C for 30 s; extension at 72 °C for 30 s; and final elongation at 72 °C for 10 min. The PCR products of 3 replicates of each sample were mixed, and 1% agarose gel electrophoresis was used to detect the quality of the PCR products. The products were mixed in equidensity ratios according to GeneTools analysis software (version 4.03.05.0, SynGene). An EZNA Gel Extraction Kit (Omega, USA) was used for purification of the mixed PCR products.

Library preparation and sequencing

Libraries were processed with the NEBNext® Ultra™ DNA Library Prep Kit for Illumina® (New England Biolabs, USA). The libraries were assessed using a Qubit@ 2.0 fluorometer (Thermo Scientific) and an Agilent Bioanalyzer 2100 system, and sequenced on the Illumina HiSeq 2500 platform to generate 250-bp paired-end reads. High-quality clean reads were obtained following the Trimmomatic quality control process (V0.33, http://www.usadellab.org/cms/?page=trimmomatic). The clean paired-end reads from which the barcodes and primers were removed, were merged using FLASH (V1.2.11, https://ccb.jhu.edu/software/FLASH/). Effective clean tags were retained after filtering the spliced sequences using Trimmomatic software.

OTU cluster and species annotation

Usearch software (V8.0.1517, http://www.drive5.com/usearch/) was applied for sequence analysis. Sequences with ≥97% similarity were assigned to the same operational taxonomic unit (OTU). Singletons were removed using usearch (http://www.drive5.com/usearch/manual/chimera_formation.html) after OTU clustering, and the chimera sequences were detected and removed using the UCHIME de novo algorithm (http://www.drive5.com/usearch/manual/uchime_algo.html). The Greengenes (http://greengenes.secondgenome.com/) and UNITE V8.0 databases were consulted based on the RDP Classifier algorithm and the assign_taxonomy.py script (http://qiime.org/scripts/assign_taxonomy.html) in QIIME for taxonomic annotation (the confidence threshold was set to 0.8). OTUs that were annotated as chloroplasts or mitochondria (16S rRNA) and could not be annotated at the kingdom level were removed.

Data analysis

The Shannon index of the endophytic bacterial and fungal communities was calculated using QIIME (V1.9.1), and the Kruskal-Wallis test at the P < 0.05 level was used to examine the effects of elevation and species on the diversity of endophytic bacterial and fungal community. Nonmetric multidimensional scaling (NMDS) analysis based on Bray-Curtis distance was performed to visualize the dissimilarities of endophytic bacterial and fungal community composition in different species and at different elevations. Permutational multivariate analysis of variance (PerMANOVA) using the “Anosim” function in R (3.6.1) with 999 random permutations was performed to compare differences in endophytic community structure between plant species and elevations. The correlations of elevation and the relative abundances of the top 20 genera were analyzed using the Spearman method. Samples from different elevations were combined for further analysis of the differences in endophytic bacterial and fungal communities between the two plant species. The linear discriminant analysis (LDA) effect size (LEfSe) method with a threshold of 4 was used to investigate indicator species that exhibited significant differences in relative abundance between plant species. KEGG functional analysis of endophytic bacteria was performed using Phylogenetic Investigation of Communities by Reconstruction of Unobserved States (PICRUSt) (Langille et al., 2013). We also analyzed genes associated with cold resistance, antioxidant enzymes (catalase, peroxidase and superoxide dismutase), nitrogen metabolism (nitrogen fixation, nodulation protein, nitrogen regulatory protein, and nitrate reductase) and phosphorus metabolism (phosphatase). Ca2+ signaling (Monroy, Sarhan & Dhindsa, 1993), cold shock proteins, HSP20 family proteins (Elkelish et al., 2020), glutathione (Liu et al., 2020a) glycerol-3-phosphate acyltransferases (Gomes et al., 2000), glycine betaine (Annunziata et al., 2019; Chen & Murata, 2011), glycosyltransferases (Shi et al., 2020), protein kinases (histidine kinase, serine protein kinase, and tyrosine kinase) (Martín & Busconi, 2001), trehalose (Liu et al., 2020b), nitric oxide (Farnese et al., 2016), and antioxidant enzymes (Baier et al., 2019) have been reported to improve plant cold resistance.

Results

Diversity of endophytic bacterial and fungal communities

The rarefaction curves of the Shannon index reached a saturation plateau in all samples, which suggested that the sampling was sufficient to obtain most of the OTUs (Fig. S1). We found a higher Shannon index of the root endophytic bacterial and fungal communities in K. pygmaea than E. nutans. Elevation significantly affected the Shannon index of root endophytic bacteria in K. pygmaea, but not E. nutans (Fig. 1A). The Kruskal-Wallis test revealed that elevation had a significant effect on the Shannon index of root endophytic fungi in E. nutans (P = 0.021) but not K. pygmaea (P > 0.05). The Shannon index of endophytic fungi in E. nutans roots decreased significantly with increasing altitude, but an inconspicuous increasing trend was observed for K. pygmaea roots (Fig. 1B). There was no significant difference in the Shannon index of root endophytic fungi between K. pygmaea and E. nutans at lower elevations (3,350–3,570 m). The Shannon index of root endophytic fungi was much higher in K. pygmaea than E. nutans at higher elevations (3,680 and 3,800 m).

Figure 1 Shannon index of root endophytic bacteria (A) and fungi (B) in K. pygmaea and E. nutans at different elevations.

Averages ± SE are displayed (n = 3). Different letters indicate significant differences between elevations.

Community compositions of endophytic bacteria and fungi

The NMDS ordination (Fig. 2) and analysis of similarities (ANOSIM) (Table 1) showed that endophytic bacterial and fungal community composition in K. pygmaea was significantly different from that in E. nutans (R = 0.78, P = 0.001; R = 0.58, P = 0.001) (Figs. 2A, 2B). Elevation also obviously influenced root endophytic bacterial and fungal communities in E. nutans (R = 0.42, P = 0.001; R = 0.84, P = 0.001) (Figs. 2C, 2D) and K. pygmaea (R = 0.76, P = 0.001; R = 0.98, P = 0.001) (Figs. 2E, 2F). The root endophytic bacterial community composition in K. pygmaea was more sensitive to elevation than E. nutans, which was validated by the R value of ANOSIM (Table 1). Greater the R values indicate a more significant influence of altitude on community composition. The endophytic fungal community was more sensitive to elevation than the bacterial community for each species. Plant identity had a greater influence on the endophytic bacterial community than the fungal community.

Figure 2 Non-metric multidimensional scaling (NMDS) ordination of the community composition of endophytic bacteria and fungi based on Bray-Curtis distances.

(A) and (B) indicated fungal and bacterial community composition in E. nutans and K. pygmaea root regardless of elevations; (C) and (D) indicated fungal and bacterial community composition at different elevations in E. nutans root; (E) and (F) indicated fungal and bacterial community composition at different elevations in K. pygmaea root. Plant species were distinguished by colors, and elevations were distinguished by shapes.

Table 1 Significance test of community composition differences among elevations and hosts determined by permutational multivariate analysis of variance using distance matrices (PERMANOVA) according to Bray-Curtis distances.

			Anosim	
			R	P	
Bacteria	Elevation	E. nutans	0.416	0.001	
K. pygmaea	0.756	0.001	
Host species	E. nutans vs K. pygmaea	0.78	0.001	
Fungi	Elevation	E. nutans	0.84	0.001	
K. pygmaea	0.984	0.001	
Host species	E. nutans vs K. pygmaea	0.579	0.001	

Proteobacteria was the predominant phylum in E. nutans and K. pygmaea roots (Fig. S2A). The average relative abundance of Proteobacteria was 58% in K. pygmaea roots, which was much lower than E. nutans (80%). Actinobacteria was the subdominant phylum in K. pygmaea roots and accounted for 30% of the bacterial reads, which was much higher than its abundance in E. nutans roots (1.4%). There were significant differences in root endophytic bacterial community composition at the order level between K. pygmaea and E. nutans (Fig. 3A). For E. nutans roots, Pseudomonadales was the most abundant order, with the relative abundance ranging from 36% to 69% along the elevation gradient. The endophytic bacterial community composition was more uniform in K. pygmaea roots than E. nutans roots. Rhizobiales accounted for 20% of the total community abundance, and there was no obvious difference between elevations. Pseudomonadales was highly abundant (50%) at the elevation of 3,800 m, and Pseudonocardiales showed low abundance at this elevation.

Figure 3 Relative abundance of bacterial (A) and fungal taxa (B) at the order level.

ER E. nutans root; KR K. pygmaea root; the numbers represent elevations. For example, ER3350 indicate E. nutans root at the elevation of 3,350 m; KR3350 indicate K. pygmaea root at the elevation of 3,350 m. The same as below.

Most endophytic fungi in all samples belonged to Ascomycetes, with average 76% for E. nutans roots and 61% for K. pygmaea (Fig. S2B). Helotiales were the dominant order in E. nutans roots and were more abundant at high altitudes, which was different from that in K. pygmaea (Fig. 3B). Pleosporales were more abundant in K. pygmaea roots than E. nutans roots, especially at lower altitudes (3,350–3,570 m). Agaricales were enriched in K. pygmaea roots at higher altitudes (3,680 m and 3,800 m) and more abundant in E. nutans roots (Fig. 3B). The relative abundance of Hypocreales in K. pygmaea roots were 18.55% and 8.1% at 3 350 m and 3,460 m, respectively, which was higher than E. nutans roots at the same elevations (Fig. 3B).

We compared the abundances of the top 20 OTUs which covered more than 50% of the total reads in all samples at different altitudes and found that plants tended to be enriched with specific endophytes at different elevations (Figs. S3A– S3D). Species with significant differences in relative abundance between K. pygmaea and E. nutans were analyzed according to LEfSe (Fig. 4). The results indicated that bacterial species belonging to Actinobacteria (Frankiales, Pseudonocardiales and Micromonosporales) and Rhizobiales (Rhizobiaceae) were highly enriched in K. pygmaea roots, and Gammaproteobacteria (Burkholderiaceae), Pseudomonadales (Pseudomonadaceae) and Flavobacteriales (Flavobacteriaceae) were highly enriched in E. nutans roots (Fig. 4A). Fungal species belonging to Dothideomycetes (Pleosporales), Eurotiomycetes (Chaetothyriales), and Sordariomycetes (Coniochaetales, Hypocreales) were highly enriched in K. pygmaea roots, and Helotiales (Hyaloscyphaceae, Helotiaceae) and Tremellales (Bulleribasidiaceae) were highly enriched in E. nutans roots (Fig. 4B).

Figure 4 Cladograms of LEfSe showing bacterial and fungal taxa with significant differences in relative abundance between K. pygmaea and E. nutans.

The filled circles from inside to outside indicate the taxonomic levels with phylum, class, order, family, genus, and species. Circles or nodes shown in color corresponding to different plant species represented a significantly more abundant group. Yellow circles indicate species with no significant differences in relative abundance.

The correlations of the relative abundances of the top 20 genera and elevation were analyzed (Table 2). For bacterial communities, the relative abundances of Pantoea, Sphingomonas, and Paenibacillus in E. nutans roots showed positive correlations with elevation. The relative abundances of Pseudomonas, Pantoea, Serratia, and Paenibacillus in K. pygmaea roots was positively correlated with elevation, and Rhizobium, Acidibacter, Actinophytocola, and Cryptosporangium was negatively correlated with elevation. For fungal communities, the relative abundances of Mortierella and Cistella showed negative correlations with elevation in E. nutans roots. The relative abundances of Mycena, Mortierella, Ilyonectria, Sebacina, and Leptosphaeria showed positive correlations with elevation in K. pygmaea roots.

Table 2 Spearman correlation analysis between elevation and the relative abundance of bacterial and fungal at genus level.

	Genus	K. pygmaea	E. nutans	
		R	P	R	P	
Bacteria	Pseudomonas	0.731	0.002	0.153	0.587	
	Pantoea	0.633	0.011	0.556	0.031	
	Rhizobium	−0.546	0.035	0.011	0.969	
	Acidibacter	−0.676	0.006	−0.207	0.458	
	Actinophytocola	−0.524	0.045	0.022	0.938	
	Cryptosporangium	−0.676	0.006	−0.016	0.954	
	Serratia	0.535	0.040	0.491	0.063	
	Sphingomonas	−0.262	0.346	0.644	0.010	
	Paenibacillus	0.742	0.002	0.764	0.001	
Fungi	Mycena	1.000	0.000	−0.700	0.188	
	Mortierella	0.900	0.037	−0.900	0.037	
	Cistella	0.700	0.188	−0.900	0.037	
	Ilyonectria	0.900	0.037	−0.800	0.104	
	Sebacina	0.900	0.037	−0.700	0.188	
	Leptosphaeria	0.975	0.005	−0.616	0.269	

The results of 16S rRNA functional prediction indicated that the endophytic bacterial community in E. nutans roots had a higher abundance of genes associated with nitrogen fixation, phosphatase activity and antioxidase activity and a lower abundance of genes associated with cold resistance than K. pygmaea roots (Fig. 5). Genes associated with nutrient absorption, including nitrogen fixation, nodulation, and phosphatase activity, were enriched in K. pygmaea roots at higher elevations. These genes were enriched at lower elevations in E. nutans roots, (Fig. S4).

Figure 5 Differences in relative abundance of functional genes associated with cold resistance, nitrogen absorption, phosphatase and antioxidase between K. pygmaea and E. nutans.

Discussion

Diversity of the endophytic community along an elevation gradient

Most studies on the influence of climate change on plant associated microbial communities overlooked host responses and did not link the responses of hosts and endophytes (Qian et al., 2018; Cordier et al., 2012). Therefore, the significance of changes in the endophytic community of the host in response to climate change were not well explained. Two plant species showing contrasting responses to climate warming were selected in the present study to fill in this knowledge gap. The increase in endophytic fungal diversity in E. nutans and decrease in K. pygmaea with warming (i.e., decreasing elevation) indicated that the endophytic fungal diversity of plants growing in suitable environments was higher than plants growing in unsuitable habitats. Our findings demonstrated that changes in endophytic fungal diversity under climate warming were closely related to host adaption, and endophytic diversity decreased when the climate became unfavorable for plant growth. A previous study indicated that endophytic diversity in sugar maple seedlings was higher within its natural range than at the edge of species’ elevational range (Wallace, Laforest-Lapointe & Kembel, 2018), which was consistent with the results of our study. We hypothesized that the loss of endophytic diversity in adverse environments was related to host selection. The symbiotic relationships between the host and some endophytes break down, and some endophytes that are beneficial to host survival are selectively enriched (Werner et al., 2018).

Plant adaptability to environmental changes is very complex and affected by many factors. We just found a phenomenon that fungal diversity was higher in the habitat that suitable for host growth. However, no direct cause-and-effect relationship between fungal diversity and host adaptation was demonstrated in the present study. Whether higher endophytic diversity contributes to host growth is worthy to study in the future.

Species and elevation significantly affected the root endophytic bacterial and fungal communities

Previous studies indicated that plant endophytes showed significant host preferences (Toju, Kurokawa & Kenta, 2019; Yao et al., 2019). Different plant species sharing similar environments recruit different microbial communities in roots (Aleklett et al., 2015), which may be related to the metabolic characteristics of the host. Because root exudates contain important factors that shape endophytic microbiome assembly, such as salicylic acid (Lebeis et al., 2015) and jasmonic acid (Carvalhais et al., 2015). Environmental parameters are also key factors that cause changes in the endophytic community (Carper et al., 2018; Bei et al., 2019; Chen et al., 2019). Climate characteristics, soil nutrients and vegetation composition change along altitude gradients may cause variations in the endophytic community (Cai et al., 2020; Zarraonaindia et al., 2015). Plant species and altitude significantly affected the root endophytic bacterial and fungal community composition in the present study. The bacterial community composition was affected more strongly by plant species, and elevation more strongly affected the fungal community. Previous studies also concluded that fungal communities were more susceptible to geographic distance than bacterial communities (Coleman-Derr et al., 2016; Meiser, Bálint & Schmitt, 2014).

Among the endophytic bacteria, Proteobacteria was the predominant phylum in E. nutans and K. pygmaea roots, which was consistent with many other studies involving different plant species (Bulgarelli et al., 2012; Beckers et al., 2017; Carrell & Frank, 2015). However, K. pygmaea roots had a higher abundance of Actinobacteria than E. nutans roots. At the order level, the relative abundances of Pseudonocardiales, Frankiales and Rhizobiales in K. pygmaea roots were much higher than E. nutans roots. However, E. nutans roots had a higher abundance of Pseudomonadales. These root endophytes have been reported to promote host plant growth and help plants resist biotic and abiotic stress (Hasegawa et al., 2006; Santoyo et al., 2016). Pseudomonas participates in nitrogen fixation (Yan et al., 2008) and iodoacetamide (IAA) synthesis (Taghavi et al., 2009), which promote plant growth. In addition to symbiosis with leguminous plants, Rhizobium acting as an endophyte of nonleguminous plants promote nitrogen and phosphorus uptake and improve the photosynthetic rate (Yanni et al., 2001; Chi, 2006). Helotiales, which promote phosphorus absorption in nonmycorrhizal plants (Almario et al., 2017), were enriched in E. nutans roots. Some genera, such as Mortierella, showed positive correlations with elevation in K. pygmaea but negative correlations with elevation in E. nutans, which also reflected the host selectivity to endophytes.

Functional differences of root endophytic bacteria in K. pygmaea and E. nutans

Chilling stress is an important environmental factor that affects plant growth and geographical distribution (Lv et al., 2018), which is especially true on the Qinghai-Tibet Plateau. A previous study also demonstrated that temperature and altitude factors primarily affected the distribution of alpine meadow grass (Zhang et al., 2020). Symbiosis with bacteria that could improve the cold tolerance of the host represents a survival strategy in alpine regions with high elevations (Acuna-Rodriguez et al., 2020; Tiryaki, Aydin & Atici, 2019). Beirinckx et al. (2020) also indicated that the root microbiome promoted maize growth under chilling conditions. The present study provided new insight into the role of endophytic bacteria in host adaptation to climate warming in terms of bacterial functional genes associated with cold resistance and nutrient absorption.

The study area in the present study exhibited a cold and wet climate at high altitudes and a warm and dry climate at low altitudes. Temperature is the main factor affecting plant growth at high elevations. Increased recruitment of bacteria with cold resistance ability in K. pygmaea roots made it more adaptable to high altitude than E. nutans although the abundance of genes associated with nutrient uptake was higher in E. nutans than K. pygmaea. Because the expression of these genes and the enzymes activity involved in nutrient absorption are restricted by low temperatures. Therefore, symbiosis with high abundance of these endophytes in E. nutans cost plant carbohydrates without getting the expected return at high altitudes.

The outcomes of symbiosis with microbiomes are environmentally dependent (Rubin et al., 2020). For example, arbuscular mycorrhizal fungi significantly enhance phosphorus absorption and promote plant growth in phosphorus deficient soils, but the contribution of mycorrhizal pathways to phosphorus uptake in plants is weakened under high P soil condition (Chu et al., 2020). Climate warming relieves cold stress at high elevations and exacerbates heat and drought stress at lower elevations (Reinhardt et al., 2011; Moyes et al., 2013). With climate warming or decreasing altitude, plant growth is freed from the cold limitation. Selectively symbiosis with bacteria that have the ability to facilitate host nutrient absorption, such as nitrogen fixation and phosphorus solubilization, in E. nutans roots made this plant more adaptable to warming environments than K. pygmaea.

Endophytes do not always establish harmonious symbioses with plant hosts. They also cause host disease and other adverse effects. There in a ‘balance of antagonisms’ relationship between fungal and plant partners (Schulz & Boyle, 2005). We only focused on the “good” side of endophytes for their importance in maintaining a stable symbiotic relationship. Exploring beneficial roles of endophytes to their hosts in special environments such as the Qinghai-Tibet Plateau contributes a lot to the development and application of microbial resources. However, functional prediction alone is not sufficient to directly demonstrate the role of endophytic bacteria in host responses to climate changes. Further studies should concentrate more on the interactions between endophytic community and host to rich our understanding of responses of terrestrial ecosystem to global warming.

Figure 6 Frame diagram of response of endophytic bacterial and fungal community to altitude.

The proportion of Elymus nutans in vegetation community as well as its endophytic fungal diversity decreased with elevation increasing, while the proportion of Kobresia pygmaea as well as its endophytic fungal diversity showed a slight increase trend with elevation increasing. The Shannon index of root bacterial and fungal community in Kobresia pygmaea were higher than that in Elymus nutans. Besides, bacteria in Kobresia pygmaea root had more genes associated with cold resistance, and in Elymus nutans root had more genes associated with nutrient absorption.

Conclusions

The present study demonstrated that the diversity of the endophytic community was higher in hosts growing in habitats that were conducive to its growth. K. pygmaea, with higher endophytic diversity and greater abundance of genes associated with cold resistance, was suited for growth in cold areas at high altitudes, and its growth was suppressed by warming. Higher endophytic diversity and greater abundance of genes associated with nutrient absorption and oxidation resistance in warmer environments (lower altitude) may contribute to the growth of E. nutans under global warming (Fig. 6). We also found that the relative abundances of the same taxa in different hosts showed different correlations with elevation, which demonstrated that the ability to recruit endophytes differed between the hosts when the habitat changed. Our study highlights the importance of plant endophytes in the responses of terrestrial ecosystems to climate change.

Supplemental Information

Figure S1 The rarefaction curves of OTU richness and Shannon index of endophytic bacterial and fungal community

ER indicated Elymus nutans root, KR indicated Kobresia pygmaea root. 3,350, 3,460, 3,70, 3,680, 3,800 indicated elevations at 3,350 m, 3,460 m, 3,570 m, 3,680 m and 3,800 m, respectively. 1, 2, 3 represent repetitions.

Click here for additional data file.

Figure S2 Relative abundance of bacteria (A) and fungi (B) at the phylum level

ER represent E. nutans root; KR represent K. pygmaea root; the number represent elevations. For example, ER3350 indicate E. nutans root at the elevation of 3,350 m; KR3350 indicate K. pygmaea root at the elevation of 3,350 m. The same as below.

Click here for additional data file.

Figure S3 Heatmap depicting the relative abundance of the top 20 bacterial and fungal OTUs along elevation gradients

ER represent E. nutans root; KR represent K. pygmaea root; the number represent elevations. For example, ER3350 indicate E. nutans root at the elevation of 3,350 m; KR3350 indicate K. pygmaea root at the elevation of 3,350 m. The same as below.

Click here for additional data file.

Figure S4 Heatmap depicting the relative abundance of functional genes associated with cold resistance, nutrient absorption (nitrogen and phosphorus), and antioxidant enzymes

Click here for additional data file.

We would like to thank Lu Yu, Wan-tong Zhang for their contribution to the filed sampling; Chang-lin Xu for his help in plant species identification.

Additional Information and Declarations

Competing Interests

Author Contributions

Data Availability

The authors declare there are no competing interests.

Xiaoting Wei conceived and designed the experiments, performed the experiments, analyzed the data, prepared figures and/or tables, authored or reviewed drafts of the paper, and approved the final draft.

Fengyan Jiang and Bing Han performed the experiments, prepared figures and/or tables, authored or reviewed drafts of the paper, and approved the final draft.

Hui Zhang performed the experiments, authored or reviewed drafts of the paper, and approved the final draft.

Ding Huang conceived and designed the experiments, analyzed the data, prepared figures and/or tables, authored or reviewed drafts of the paper, and approved the final draft.

Xinqing Shao conceived and designed the experiments, analyzed the data, prepared figures and/or tables, and approved the final draft.

The following information was supplied regarding data availability:

The 16S rRNA and ITS rRNA sequences are available at NCBI SRA: PRJNA687152 and PRJNA687162.

The 16S rRNA and ITS rRNA sequences are also available at figshare: wei, Xiaoting (2021): Root endophytic bacterial community. figshare. Dataset. https://doi.org/10.6084/m9.figshare.13520078.

wei, Xiaoting (2021): Root endophytic fungal community. figshare. Dataset. https://doi.org/10.6084/m9.figshare.13520105.v1.

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
