# Peer review of "New insight into the divergent responses of plants to warming in the context of root endophytic bacterial and fungal communities"

_PeerJ, doi:10.7717/peerj.11340_

## Round 0.1 · original submission · Major Revisions

I recommend that authors take into account the comments of the reviewers to improve this work.

Reviewer 1 ·

Basic reporting

1. The manuscript uses clear and unambiguous English. However, there are some sentences that need improvement. I recommend a native English-speaker to proofread the text for minor editing, but some examples of sentences to be corrected are:

L. 26-27: The gap this study addressed is not clearly stated in the abstract. Perhaps the issue here is the structural problem of the sentence. Please, review this sentence to improve comprehension.

L. 33-35: Rewrite this sentence to improve comprehension. My suggestion is: "Both elevation and host identity significantly affected the composition of root endophytic bacterial and fungal communities."

L. 306: insert a “to” before “host”

2. The references cited in the manuscript provide the necessary background of the study.

3. Raw data were shared and it is accessible.

4. All figures are relevant for the study, but a few corrections should be carried out to improve comprehension:

Figure 1, please, add the statistics used to find differences between groups.

Figure 2, I was a bit lost regarding letters A, B, C, D etc. One suggestion is to label A and B (and so the others) in the horizontal line of the figure to improve comprehension.

Figure 4: what are the yellow circles? Please, include this info in the figure legend.

6. Structure of the manuscript:

L. 109-111: I believe this sentence fits better before introducing the model system of study (in L. 102).

L. 223-226: These sentences fit better at the beginning of the results section (L. 215).

The abstract is clearly written for the most part. I was a bit frustrated at the end of the abstract because there is no general conclusion of the study. Thus, to improve even more the abstract, a general conclusion could be added at the end of the abstract. In this conclusion, try to link it with the gap of the study.

Experimental design

The manuscript brings original results of environment effects on the diversity and composition of plant-endophytic communities. The research question is well defined and fills the gap in the current literature (i. e. linking adaptations of host plant to environmental stress with changes in root-endophyte communities. In order to help readers understanding the experimental set up, please consider the following:

L. 129: How many plants were sampled per plot? Were the data of different plant species averaged considering the same plot? I did not find the actual number of samples (roots and plant individuals) analyzed in this study. Because of that, interpretations of figures and tables are somewhat difficult. It appears from the text that three samples were analyzed at each altitude per plant species. This should be clear stated in the manuscript.

Also, why choose the 20 TOP OTUs and not the 10 TOP OTUs? What was the criteria for this selection?

Validity of the findings

All underline data was provided. The conclusions are well stated. My only concern with the conclusions is how the sampled sized used in the experiments is sufficient to support the conclusions. See my comments above about the sample size.

Additional comments

This is an interesting manuscript that explored the association of host environmental adaptation to stress conditions and changes in endophytic microbial communities. I commend the authors to explore this link, because this is indeed a gap in the literature. When reading this manuscript, I had the following thoughts that I think it is worth to bring and I hope they can be useful for strengthening the message of this manuscript.

1. In their 2005 review, Barbara Schulz and Christine Boyle state that endophytes do not necessary establish harmonious symbioses with plant hosts. Is there a particular reason to focus only on the "good" side of endophytes? Please, check the review in 10.1017/s095375620500273x

2. L. 61-62: Are the endophytic communities responding to the plant stress or are the plants responding to the changes of endophytic communities?

3. The manuscript brings that “…greater abundance of genes associated with nutrient absorption and oxidation resistance in warmer environments (lower altitude) might contribute to the growth of E. nutans under global warming (Fig. 6).”

I would suspect the genes “associated with nutrient absorption and oxidation resistance” should be abundant not only in E. nutans but also in K. pygmaea bacterial microbiome, because they are also necessary to survive in cold conditions.

·

Basic reporting

Wei et al. studied the response of two plant species and their endophytes in relation to warming condition. Authors studied both bacteria and fungi. The article is generally well written.

I feel there is still some scope for the improvement of the manuscript. My suggestions are following in other sections.

Experimental design

Authors selection of two plant species is good enough to answer their research question. The methods used are very standard in the plant endophytes research.

Authors have not mentioned that how many plants (roots) were sampled and how many of them were sequenced. Number of samples per plot or per plant species and total number samples should be clearly explained in the materials and methods section.

Validity of the findings

I highly appreciate that the authors shared the raw data of this study. The standard statistical methods are used in the manuscript.

Additional comments

Line 111 to 114: The introduction is more general. I feel detailed explanation of their hypothesis citing relevant literatures is necessary to make the readers understand the scope the of the work well. The different plant species can have different endophytes just because of their physiology. Please explain it in detail still how do you prove your hypothesis in that case.

Line 150, 186 and in abstract: Please use the terms same throughout the manuscript. For example, 16S rRNA gene,16S rDNA and 16S amplicons are used in the manuscript. It should be corrected as 16S rRNA gene through out the manuscript.

Line 150 to 154: Authors should cite the publication relevant to the primers they have used.

Line 150 to 151: I hope the 16S rRNA gene PCR amplification results in mitochondrial as well as bacterial amplification. Could you give some details about how did you overcome this problem?

Line 157: ......RNA amplification....? please correct it. It is not clear.

Line 297 to 299: How the results consistent with your hypothesis? It is just partially consistent as the bacterial diversity is different? Change in fungal diversity does not mean that plants are adopted due to that. Please give more explantation.

Reviewer 3 ·

Basic reporting

See below.

Experimental design

See below.

Validity of the findings

See below.

Additional comments

This paper has analyzed whether and how elevation influences fungal and bacterial composition in two contrast plant species, one more adapted to high elevation (K. Pygmaea), the other more adapted to low elevation (E. Nutans ) in Qinghai-Tibet Plateau.
ITS and 16S meta-sequencing analysis conducted in this study suggest the following results.
1. Plant genotype influences both fungal and bacterial composition. This seems to be consistent with the previous studies.
2. Diversity of fungal communities in E. Nutans significantly decreases in a negative correlation with elevation. This might be because E. Nutans grow poorly in high elevation.
3. Fungal or bacterial species specifically enriched in either plant species are identified.
4. K. Pygmaea associates with bacteria with genes related to Cold resistance. In contrast, E. Nutans associates with bacteria with genes related to nutrient acquisition. This seems to be reasonable results considering their habitat ( K. Pygmaea more grows in colder elevation than in E. Nutans).

Over all, this paper contains useful information for researchers studying microbiome and ecology. The method itself seems to be approbate. However, due to poor English throughout the manuscript, I felt difficulties to catch the contents. It is essential for the authors to submit this paper to a professonal English editing service.

Minor points

: In material and method, I don’t fully catch how the authors collected the samples from two different plant species grown in various elevation. It is necessary to show this in a single picture. I think modifying figure 6 would be useful for this purpose.
: Please describe how many plant samples are used per one conditions.

---

## Round 0.2 · Minor Revisions

The authors should consider the minor comments of reviewer 3.

·

Basic reporting

Authors improved the manuscript in response to all the reviewers comments.

Experimental design

The experimental design is very standard in the manuscript.

Validity of the findings

All the data are available.

Additional comments

Thank you for your response to my comments.

Reviewer 3 ·

Basic reporting

This version is significantly improved. I have only pointed out a minor point as described below.

Experimental design

Looks fine.

Validity of the findings

This should be fine.

Additional comments

The authors have mentioned that pants belonging to Brassicaceae recruit another fungus, Helotiales, under phosphorus-limited conditions (Almario et al., 2017). To mention this, it would not be appropriate to cite Hiruma et al., 2016 that show that A. thaliana recruit an endophytic fungus Colletotrichum tofieldiae, under phosphorus-limited conditions.
Although it is not essential, this paper would be significantly improved if the authors have cited appropriate papers related to plant-microbe interactions.

---

## Round 0.3 · accepted · Accept

All comments of the reviewers were taken into account in the new version of the manuscript. I believe the manuscript is ready for publication.